# A CRISPR Screen of HIV Dependency Factors Reveals That *CCNT1* Is Non-Essential in T Cells but Required for HIV-1 Reactivation from Latency

**DOI:** 10.3390/v15091863

**Published:** 2023-08-31

**Authors:** Terry L. Hafer, Abby Felton, Yennifer Delgado, Harini Srinivasan, Michael Emerman

**Affiliations:** 1Molecular and Cellular Biology Graduate Program, University of Washington, Seattle, WA 98195, USA; thafer@fredhutch.org; 2Divisions of Human Biology and Basic Sciences, Fred Hutchinson Cancer Center, Seattle, WA 98109, USA; 3Bioinformatics Shared Resource, Fred Hutchinson Cancer Center, Seattle, WA 98109, USA

**Keywords:** HIV latency, dependency factors, P-TEFb complex, latency reversal agent (LRA), CRISPR screening, Cyclin T1, T cells, CCNT1

## Abstract

We sought to explore the hypothesis that host factors required for HIV-1 replication also play a role in latency reversal. Using a CRISPR gene library of putative HIV dependency factors, we performed a screen to identify genes required for latency reactivation. We identified several HIV-1 dependency factors that play a key role in HIV-1 latency reactivation including ELL, UBE2M, TBL1XR1, HDAC3, AMBRA1, and ALYREF. The knockout of Cyclin T1 (*CCNT1*), a component of the P-TEFb complex that is important for transcription elongation, was the top hit in the screen and had the largest effect on HIV latency reversal with a wide variety of latency reversal agents. Moreover, *CCNT1* knockout prevents latency reactivation in a primary CD4+ T cell model of HIV latency without affecting the activation of these cells. RNA sequencing data showed that CCNT1 regulates HIV-1 proviral genes to a larger extent than any other host gene and had no significant effects on RNA transcripts in primary T cells after activation. We conclude that CCNT1 function is non-essential in T cells but is absolutely required for HIV latency reversal.

## 1. Introduction

The existence of an activatable latent reservoir is a key barrier to virus elimination in people living with HIV as cells harbor an integrated latent proviral genome that persists in the presence of antiretroviral treatment. The multifaceted nature of HIV latency suggests a combination of methods and approaches will need to be used to effectively reduce this reservoir. Factors that ultimately block HIV-1 transcription including host epigenetic silencing mechanisms, blocks to transcription initiation, and transcription elongation all contribute to a silent, or nearly silent, HIV reservoir.

The “shock and kill” approach to reservoir reduction involves using latency reversal agents (LRAs) to promote viral transcription and viral reactivation in the latent reservoir and then eliminating those reactivated cells using immunological approaches or methods that rely on the recognition of newly synthesized viral proteins [1,2,3]. The shock and kill approach is attractive in that it seeks to eliminate the latent reservoir by killing cells harboring transcriptionally competent proviral sequences. However, these LRAs must target a broad range of proviruses with highly variable epigenetic and gene expression contexts in different cells and tissues [4,5]. Another strategy, called “block and lock”, involves targeting factors that are required for HIV replication in order to prevent viral reactivation [6,7]. Such approaches rely on molecules called Latency Promoting Agents (LPAs) that seek to lock the HIV promoter into a permanently silenced state. For instance, didehydro-Cortistatin A (dCA) inhibits Tat/TAR interaction and, therefore, enforces latency by inhibiting Tat transactivation [8]. Other approaches have used siRNAs to target the LTR and prevent transcription of proviral genes, which can lead to epigenetic silencing on the recruitment of histone-modifying complexes to the LTR region [9,10]. Thus far, only one block-and-lock drug, ruxolitinib, a JAK/STAT inhibitor, has made it to a clinical Phase 2a study [11]. Both “shock and kill” and “block and lock” therapeutic approaches will likely involve manipulation of multiple arms of HIV latency for a desired outcome, and, therefore, a more comprehensive understanding of these mechanisms is an important consideration for approaches to eliminate the latent reservoir and achieve a functional HIV cure.

We previously performed a CRISPR screen using a novel system called Latency HIV-CRISPR to identify host genes involved in epigenetic control that maintain latency [12]. In this screen, the knockout of genes promotes reactivation from latency, suggesting that these host genes normally function to repress HIV-1 transcriptional activation. In the present study, we modified this system to identify host genes that are required for HIV-1 to reactivate from latency, i.e., necessary for HIV-1 to come out of latency. We hypothesized that a subset of host genes that HIV requires for replication, called HIV dependency factors, would also be required for reactivation from latency. Our goal was to identify proteins whose function is more important for HIV-1 reactivation than normal T cell biology. 

Transcriptional regulation of HIV-1 is dependent on several host mechanisms, with the P-TEFb complex being a key component that interacts with a viral protein, Tat, and a viral RNA element, TAR, to allow for transcription elongation. Both HIV-1 and host genes use CCNT1 and CDK9 in the P-TEFb complex in order to enable transcription elongation [13]. CCNT1 has a paralog—CCNT2—which also forms the P-TEFb complex [14], and in vitro studies have shown that another host protein, CCNK, can also interact with CDK9 to form the P-TEFb complex [15]. However, while HIV-1 Tat viral protein binding sites are conserved in CCNT1 and CCNT2, only the CCNT1-Tat complex can bind with the viral TAR RNA in order to recruit P-TEFb to the LTR [16].

Here, we performed a CRISPR-Cas9 screen using the Latency HIV-CRISPR technique [12] for factors necessary for HIV-1 to be released from latency in the presence of a combination of LRAs. We used a custom CRISPR guide library, called the HIV dependency factor gene library (HIV-Dep), that had been previously used to identify novel host dependency factors across multiple HIV strains [17]. We identified and validated factors important in latency reactivation including ELL1, TBL1XR1, UBE2M, HDAC3, AMBRA1, and ALYREF. Cyclin T1 (*CCNT1*), which forms the P-TEFb transcriptional elongation complex with Cyclin-dependent Kinase 9 (CDK9), was the top hit in two J-Lat models in our screen. We found that Cyclin T1 is essential for reactivation from latency in J-Lat cells as well as in a primary T cell model of HIV latency using a broad range of LRAs. *CCNT1* knockout had no effect on cell proliferation in the J-Lat model and did not affect activation through the T cell receptor in primary CD4+ T cells. Moreover, we performed bulk RNA sequencing on *CCNT1* knockouts and found that HIV-1 genes were the most depleted relative to wild type *CCNT1* over any host gene in J-Lat cells, whether or not they were treated with an LRA. RNA sequencing in uninfected primary T cells knocked out for *CCNT1* showed very few changes in host cell transcript expression. Together, our findings show that some HIV-1 dependency factors are more important for HIV replication and reactivation than host cell biology and suggest that *CCNT1* could be a promising therapeutic target for silencing HIV-1 into deeper latency. To that end, other genes uncovered in our screen may also be worth exploring further as factors for a block and lock mechanism for HIV.

## 2. Methods

### 2.1. Cell Culture and Maintenance

HEK293T cells were cultured in DMEM (ThermoFisher, Grand Island, NY, USA, 11965092) along with Penicillin/Streptomycin (Pen/Strep) and 10% Fetal Bovine Serum (FBS). J-Lat cells were cultured in RPMI 1640 media (ThermoFisher, Grand Island, NY, USA, 11875093) supplemented with Pen/Strep, 10% Fetal Bovine Serum (FBS), and 10 mM HEPES (ThermoFisher, Scotland, UK, 15630080). Cells were maintained at 37 °C with 5% CO_2_. Cells were routinely tested and found to be free of mycoplasma contamination. The primary CD4+ T cell media used was RPMI 1640 + 1× Anti-Anti (Gibco, Grand Island, NY, USA, 15240096), 1× GlutaMAX (ThermoFisher Scientific, Grand Island, NY, USA; 35050061), 10 mM HEPES, and 10% FBS.

### 2.2. HIV-CRISPR Library Transduction and Virus-Encapsidated CRISPR Guide Screening

The HIV-Dep library containing 525 genes (4191 sgRNAs) was previously described [17]. For the transduction of J-Lat cells, HEK293T cells were seeded in 20 × 6 well cell culture plates, transfected with the HIV-DEP plasmid (667 ng), psPax2 (GagPol, Addgene, 12260; 500 ng), and MD2.G (VSVG, Addgene, 12259; 333 ng) per well in 200 μL of serum-free DMEM (Thermo Fisher Scientific, Grand Island, NY, USA, 11965092) along with 4.5 μL of TransIT-LT1 reagent (Mirus Bio LLC, Madison, WI, USA, MIR2305). VSVG pseudotyped lentivirus was harvested and filtered through a 0.22 um filter (Sigma-Aldrich, St. Louis, MO, USA, SE1M179M6). The virus was titered using TZM-bl (NIH AIDS Reagent Program; ARP-8129) cells. J-Lat 10.6 and J-Lat 5A8 previously knocked out for *ZAP* [12] were transduced with the HIV-CRISPR library lentivirus with DEAE-Dextran (final concentration 20 μg/mL, Sigma-Aldrich; Milwaukee, WI, USA D9885) at a multiplicity of infection (MOI) of 0.5. After 24 h, puromycin (Sigma, Milwaukee, WI, USA P8833) at a final concentration of 0.4 μg/mL was added to the culture to select the cells that received the vector. The screen was performed 11 days after transduction by treating the HIV-Dep library-transduced J-Lat cells with latency reversal agents AZD5582 1 nM (MedChemExpress, Monmouth Junction, NJ, USA, HY-12600) and I-BET151 2.5 μM (SelleckChem, Houston, TX, USA, S2780) or DMSO (Sigma, Milwaukee, WI, USA, 472301) control. After 24 h (day 12), the supernatants were harvested, filtered (Millipore Sigma, St. Louis, MO, USA, SE1M179M6), and loaded over a 20% sterile sucrose solution (20% sucrose, 1 mM EDTA, 20 mM HEPES, 100 mM NaCl, distilled water) placed on a prechilled SW32Ti rotor. The viral pellets were then concentrated at 70,000× *g* for 1 h at 4 °C and gently resuspended in 140 μL of DPBS (Gibco; 14190144) and allowed to resuspend overnight at 4 °C. Simultaneously, transduced cells were harvested to isolate genomic DNA (gDNA). Cells were centrifuged and resuspended in DPBS. Cells were then spun down, the supernatant was removed, and cell pellets were frozen until ready for gDNA extraction. 

### 2.3. Latency HIV-CRISPR Screen 

Viral RNA (vRNA) and gDNA were isolated as previously described [18]. Briefly, vRNA was isolated using the QIAamp Viral RNA Mini Kit (Qiagen, Germantown, MD, USA, 52904). A reverse transcription of vRNA was performed using the SuperScript Reverse Transcriptase Kit (ThermoFisher, Carlsbad, CA, USA, 18064014). gDNA was isolated using the QIAamp DNA Blood Midi Kit (Qiagen, Germantown, MD, USA, 51183). vRNA and gDNA were both amplified by PCR using R1_forward primer and R1_Reverse primer using Herculase II Fusion DNA Polymerase (Agilent, Santa Clara, CA, USA, 600677). PCR products were cleaned up using the QIAquick PCR clean-up kit (Qiagen, Germantown, MD, USA, 28104), and a second round of PCR was performed using R2_reverse primer and R2_IndexX primer (described in [12,17,19]). The 230 bp band was verified to be present and the amplified PCR products were cleaned up using double-sided SPRI via AMPure Beads (Beckman Coulter, Indianapolis, IN, USA, A63880). Purified samples were normalized to a concentration of 10 nM using Qubit dsDNA HS Assay Kit (Invitrogen, Eugene, OR, USA, Q32854) before sequencing. 

Adapter sequences were computationally trimmed from sequencing results, and the viral sequencing was compared relative to the genomic knockout pool to determine the relative enrichment or depletion of each guide. An artificial NTC sgRNA gene set was generated, which is equivalent to the number of genes present in the HIV-Dep library “synNTCs”, by iteratively binning the NTC sgRNA sequences. MAGEcK and MAGEcK Flute statistical [20,21] analyses were used to analyze the depletion of guides/genes in the RNA viral supernatant relative to their abundance in the cell DNA. Z-scores were determined as previously described [17,22]. For each HIV-Dep LAI replicate, and for each replicate of the J-Lat CRISPR screen, z-scores were calculated. An average of the z-scores from each replicate was used to generate a heatmap. Heatmaps were generated using Morpheus https://software.broadinstitute.org/morpheus (accessed on 15 May 2023). The code for z-score analysis of CRISPR screen data can be found at https://github.com/amcolash/hiv-crispr-zscore-analysis (accessed on 16 January 2023). 

### 2.4. Validation of Screen Hits

Genes identified in the HIV-Latency screen that were depleted after LRA treatment were validated either by lentiviral knockout or by the electroporation of RNA guides and Cas9. For genes validated by lentiviral knockout, a forward and reverse primer corresponding with 2 individual guides targeting each gene was cloned into pLCV2, and cells were transduced as described above. Puromycin selection continued for 10–14 days until treated with LRAs. For pooled electroporation knockout experiments, CRISPR/Cas9-mediated knockout was performed against genes of interest using Gene Knockout Kit v2 (Synthego, Redwood City, CA, USA). Guides targeting genes of interest with 1 μL of 20 μM Cas9-NLS (UC Berkeley Macro Lab, Berkeley, CA, USA) and RNP complexes were made with the SE Cell Line 96-well Nucleofector Kit (Lonza, Walkersville, MD, USA, V4SC-1096). Complexes were incubated at room temperature for ten minutes, and 2 × 10^5^ cells of J-Lat 10.6 were centrifuged at 100× *g* for 10 min at 25 °C and were resuspended in Cas9-RNP complexes and electroporated on Lonza 4D-Nucleofector using code CL-120. Cells were recovered with RPMI media pre-warmed to 37 °C. Knockout pools were maintained for 10–14 days to allow for expansion and subsequently treated with LRAs. In both cases, reactivation was measured by RT activity as described [23]. After 24 h of LRA treatment, genomic DNA was analyzed to assess the degree of gene knockouts.

For *CCNT1* knockout clones, CRISPR/Cas9-mediated knockout was performed using Gene Knockout Kit v2 (Synthego, Redwood City, CA, USA). Guides targeting *CCNT1* were complexed with 1 μL of 20 μM Cas9-NLS (UC Berkeley Macro Lab, Berkeley, CA, USA), and RNP complexes were made with the SE Cell Line 96-well Nucleofector Kit (Lonza, Walkersville, MD, USA, V4SC-1096). Complexes were incubated at room temperature for ten minutes, and 2 × 10^5^ cells of J-Lat 10.6 were centrifuged at 100× *g* for 10 min at 25 °C and were resuspended in Cas9-RNP complexes and electroporated on Lonza 4D-Nucleofector using code CL-120. Cells were recovered with media pre-warmed to 37 °C. Five days post-electroporation, single cells were sorted into a 96-well U-bottom plate filled with 100 μL RPMI media (20% FBS).

To assess the growth of *CCNT1* knockout J-Lat 10.6 relative to wild type, three individual flasks of either wild type, *CCNT1* Knockout 1, or *CCNT1* Knockout 2 J-Lat 10.6 were maintained for each line. Cells were resuspended at a concentration of 2 × 10^5^ cells/mL in a total of 10 mL RPMI media. Cells were monitored and split approximately every two days. Cell counts prior to splitting were taken and the volume of cell suspension removed (the same volume was removed for each line) was tracked and subtracted from the overall cell count. These values were tracked over a span of nine days.

### 2.5. Protein Isolation and Western Blotting

Cell pellets (1.5 × 10^6^–3 × 10^6^ cells) from pooled lentiviral knockout experiments (NTC10 and *CCNT1* sg1 and sg2) and clonal knockout experiments (J-Lat 10.6 *CCNT1* KO clone 1 and 2) were isolated from each respective experiment. The supernatant was removed, and the cells were resuspended in 500 μL of cold (4 °C) 1× PBS. Cells were pelleted, resuspended in 100 μL of RIPA buffer (150 mM NaCl (Sigma, S3014), 50mM Tris pH 8.0, 1% NP-40 (Calbiochem, St. Louis, MO, USA, 492016), 0.5% Sodium Deoxycholate (Sigma-Aldrich, St. Louis, MO, USA, D6750), 0.1% SDS (Sigma-Aldrich, St. Louis, MO, L4509), Benzonase 1 μL/mL (Millipore, 70664), and cOmplete Protease Inhibitor Cocktail (Roche, Manheim, Germany, 11697498001), and incubated on ice for 10 min with repeated vortexing. Cell lysate was pelleted at 20,000× *g* for 20 min at 4 °C. Clarified supernatant was transferred to a new tube and quantified by BCA. Samples were prepared by adding 4× NuPAGE LDS Sample Buffer (ThermoFisher, NP0007) with 5% 2-Mercaptoethanol (Sigma-Aldrich, M3148) and denatured at 95 °C for 5 min. Lysates were run on a NuPAGE 4–12% Bis-Tris pre-cast gel (ThermoFisher Scientific; NP0336) and transferred to a nitrocellulose membrane (Biorad, Hercules, CA, USA, 1620115). After transfer, the nitrocellulose membrane was blocked in a 0.1% Tween/5% Milk in 1× PBS solution for 30 min at room temperature. Primary antibodies used for western blotting were mouse α-CCNT1 (Santa Cruz Biotechnology, Dallas, TX, USA, sc-271348, 1:500), mouse α-CCNT2 (Santa Cruz Biotechnology, sc-81243, 1:500), and rabbit α-actin (Sigma-Aldrich, A2066 1:5000). Antibodies were diluted in 1× PBS-Tween 0.1% (PBST) and rocked on nitrocellulose membrane overnight at 4 °C. The membrane was washed with PBST 3–5 times for 5 min each wash. The following secondary antibody dilutions were made at 1:2000 in PBST: goat α-mouse IgG-HRP (R&D Systems, Minneapolis, MN, USA, HAF007) and goat α-rabbit IgG-HRP (R&D Systems; HAF008). SuperSignal West Femto Maximum Sensitivity Substrate (ThermoFisher; 34095) was used for CCNT1 and CCNT2 and SuperSignal West Pico PLUS Chemiluminescent Substrate (ThermoFisher, 34580) was used for Actin. Visualization was performed on a BioRad Chemidoc MP Imaging System (Biorad, Hercules, CA, USA).

### 2.6. Genomic Editing Analysis

Cells for each knockout were pelleted and washed with 1× PBS, the supernatant was removed, and cell pellets were frozen at −80 °C until ready for DNA isolation. Genomic DNA was isolated using a QIAamp DNA Blood Mini Kit (Qiagen; 51104). The gene of interest was amplified using primers described using either Q5 High-Fidelity DNA polymerase (NEB, Ipswich, MA, USA, M0491S) or Platinum Taq DNA polymerase High Fidelity (ThermoFisher Scientific; 11304011). PCR products were purified using AMPure beads (Beckman Coulter, A63880) or a QIAquick PCR clean-up kit (Qiagen, 28104) and submitted to Fred Hutch Genomics shared resource for sequencing. Analysis was performed using Inference of CRISPR Edits (ICE) [24]. Briefly, ICE analysis compares Sanger sequencing from wild type and CRISPR-edited sequences, determining the insertion and deletions from these sequences and generating knockout scores along with a correlation value as assessments of the knockout (Appendix A). 

### 2.7. LRA Treatments

For J-Lat 10.6 or J-Lat 5A8 cells, LRAs were used at the following concentrations: TNFα (Peprotech, Cranbury, NJ, USA, 300-01A) 10 ng/mL; AZD5582 (MedChemExpress, Monmouth Junction, NJ, USA, HY-12600) 1 nM; I-BET151 (SelleckChem, Houston, TX, USA, S2780) 2.5 μM; Prostratin (Sigma-Aldrich, P0077) 0.1 μM; and SAHA/Vorinostat (SelleckChem, Houston, TX, USA, S1047), 2.5 μM. For CD3/CD28 antibody stimulation, Anti-CD3 clone UCHT1 (Tonbo, San Diego, CA, USA, 40-0038-U500) was plated on a 96-well flat bottom plate at 10 μg/mL in 1× PBS, incubated overnight at 4 °C, aspirated, and CD28 clone 28.2 antibody (Tonbo, 40-0289-U500) was added to RPMI media at a concentration of 4 μg/mL for cell resuspension. Cells for each experiment were resuspended at a concentration of 5E5 cells/mL in appropriate LRA media, and 200 μL was aliquoted into a 96-well flat bottom TC plate. For Primary CD4+ T Cell LRA treatment, PMA (Sigma-Aldrich, P1585) was used at a concentration of 10 nM in combination with ionomycin (Sigma-Aldrich, I0634), which was used at a concentration of 1 μM. For primary cell experiments, CD3 antibody (Tonbo, 40-0038-U500) was used at a concentration of 10 μg/mL and CD28 antibody (Tonbo, 40-0289-U500) at a concentration of 5 μg/mL. All LRA treatments were performed for 24 h, unless otherwise indicated.

### 2.8. Primary CD4+ Cell Isolation and Latency Model

All centrifugation steps of Primary CD4+ T cells were performed at 300× *g* for 10 min at 25 °C unless otherwise noted. PBMCs were isolated from used leukocyte filters (Bloodworks Northwest, Seattle, WA, USA) over a Ficoll gradient (Millipore Sigma, St. Louis, MO, USA, GE17-1440-02), cryofrozen at a concentration of 10–20 × 10^6^ cells/mL in 90% FBS/10% DMSO and stored in liquid nitrogen until ready to use. On thawing, PBMCs were washed dropwise with pre-warmed RPMI-1640 media (Thermo Fisher) and treated with benzonase (25 U/mL) (Sigma-Aldrich, E1014) for 15 min at room temperature. PBMCs were maintained at a concentration of 2 × 10^6^ cells/mL overnight at 37 °C. The following day, CD4+ T cells were isolated using the EasySep Human CD4+ T cell Isolation Kit (Stemcell Technologies, Burnaby, BC, Canada, 17952) and subsequently activated using the T Cell Activation/Expansion Kit (Miltenyi Biotec, San Jose, CA, USA, 130-091-441). From this point forward, CD4+ T cells were cultured in RPMI + IL-2 (final concentration 100 U/mL, Roche, 10799068001), IL-7 (final conc. 2 ng/mL, Peprotech, Cranbury, NJ, USA, 200-07), and IL-15 (final conc. 2 ng/mL, Peprotech, 200-15) unless otherwise noted. Cells were activated continually for two days prior to infection. 

The lentivirus for the infection of primary CD4+ T cells was generated by transfecting HEK293T cells with Δ6-dGFP-Thy1.2-Gagpol+ Plasmid (900 ng, gift from Ed Browne Lab), psPax2 plasmid (450 ng), and MD2.Cocal plasmid (150 ng, gift from Hans-Peter Kiem Lab [25]. After two days, the virus was filtered using a millipore filter (Millipore Sigma, SE1M179M6).

On the day of infection, activation beads were first magnetically removed. Infection of CD4+ T cells was performed by aliquoting 5 × 10^6^ CD4+ T cells iteratively into 50 mL falcon tubes and resuspending in virus + polybrene (final conc 8 μg/mL, Sigma-Aldrich, TR-1003) or RPMI media + polybrene for the uninfected control at a concentration of 1 × 10^6^ cells/mL. Spinoculation was performed for 1100× *g* for 2 h at 30 °C. Cells were maintained at a concentration of 1 × 10^6^ cells/mL.

Three days post-infection, a small portion of cells were taken to assess infection by staining with CD90-AF700 antibody (Biolegend, San Diego, CA, USA, 140323) for 20 min (1:1000 dilution in FACS Buffer), fixing with 4% paraformaldehyde and sorting by AF700 and GFP on SP Celesta 2 Cell Analysis Machine (Flow Cytometry Core, Fred Hutch, Seattle, WA, USA). CD90+ cells were then isolated using the CD90.2 Cell Isolation Kit (Stemcell Technologies, Burnaby, BC, Canada, 18951). Two days after CD90+ cells were purified, cells then were electroporated using electroporation code EH-100 and the P3 Primary Cell 96-well Nucleofector Kit (Lonza, V4SP-3096). Knockout pools were maintained for an additional nine days prior to coculturing with an H80 feeder cell line with IL-2 (Final conc 20 U/mL) in RPMI (no longer cultured with IL-7 and IL-15). Four days later, the cells were treated with PMAi or CD3/CD28 antibody co-stimulation (or unstimulated control) and analyzed on SP Celesta 2 (Flow Cytometry Core, Fred Hutch, Seattle, WA, USA) to evaluate reactivation potential by assessing Thy1.2, CD90+, and GFP+ cells. An early activation marker of T cells was also monitored using a PE-Conjugated CD-69 antibody (Biolegend, 310906). Analysis was performed with FlowJo software, version 10.8.1. Genomic DNA was isolated at the end of the experiment from uninfected and knockout cells to assess the genomic ICE analysis.

### 2.9. Primary CD4+ T Cell Activation Test

CD4+ cells were isolated from healthy donors and activated as described above. After two days of activation, the beads were magnetically removed. Three days later, cells were electroporated following the protocol above and treated with CD3/CD28 antibody after cells were allowed to recover for two additional days. Activation was monitored using PE-Conjugated CD69 antibody (Biolegend, 310906) on SP Celesta 2. Genomic DNA was isolated for analysis.

### 2.10. RNA-seq Analysis of CCNT1 Knockout Cells

For RNA isolated from J-Lat 10.6 cells, cells first were passaged and split equally three times prior to isolation. Either J-Lat 10.6 or wild type for *CCNT1* or knocked out for *CCNT1* were each treated with TNFα (Peprotech, 300-01A) at 10 ng/mL or unstimulated in triplicate. For primary cell experiments, knockouts were performed similar to the “Primary CD4+ T cell activation Test,” and RNA was isolated after LRA treatment. In both J-Lat and primary CD4+ T cell isolation experiments, 0.1–2 × 10^6^ cells were isolated and resuspended in 350 μL of RLT Plus (Qiagen, 1053393) + 1% 2-mercaptoethanol (Millipore Sigma, M3148). Cells were frozen in buffer RLT plus until RNA isolation. Thawed RLT lysates were then run over a QIAshredder column (Qiagen, 79654) and subsequently a gDNA eliminator column. A Qiagen RNeasy Plus Mini Kit was then used in order to obtain purified total RNA. RNA was submitted for TapeStation RNA assay or HighSense RNA assay (Fred Hutch Core Facilities Seattle, Washington, DC, USA), and RINe scores were all found to be ≥9.6. 

### 2.11. RNAseq Analysis Methods 

Quality assessment of the raw sequencing data, in Fastq format, was performed with fastp v0.20.0 [26] to ensure that data had high base call quality, expected GC content for RNA-seq, and no overrepresented contaminating sequences. No reads or individual bases were removed during this assessment step. The fastq files were aligned to the UCSC human hg38 reference assembly using STAR v2.7.7 [27]. STAR was run with the parameter “--quantMode GeneCounts” to produce a table of raw gene-level counts with respect to annotations from human GENCODE build v38. To account for unstranded library preparation, only unstranded counts from the table were retained for further analysis. The quality of the alignments was evaluated using RSeQC v3.0.0 [28], including an assessment of bam statistics, read-pair inner distance, and read distribution. Differential expression analysis was performed with edgeR v3.36.0 [29] to identify the differences between knockout stimulated and stimulated for with *CCNT1* and *AAVS1* genes, as well as differences between the two genes in knockout and knockout stimulated conditions. Genes with very low expression across all samples were flagged for removal by filterbyExpr, and TMM normalization was applied with calcNormFactors to account for differences in library composition and sequencing depth. We constructed a design matrix to incorporate potential batch effects related to donor information, after which the dispersion of expression values was estimated using estimateDisp. Testing for each gene was then performed with the QL F-test framework using glmQLFTest, which outputs a *p*-value, a log2(fold change) value, and a Benjamini–Hochberg corrected false discovery rate (FDR) to control for multiple testing for each gene. The results were plotted using ggplot2 v3.3.5 [30]. For analysis of J-Lat 10.6 RNA sequencing data, we used the reference genome previously assembled and described for J-Lat 10.6 [12]. Using this reference, we masked the 5′ LTR of the integrated provirus. All splice variants, as well as genomic RNA, which terminate at a polyA site in the 3′ LTR, are similarly named “HIV-1.”

## 3. Results

### 3.1. A Latency HIV-CRISPR Screen of HIV Dependency Factors to Identify Latency Reversal Factors

We recently developed and validated a CRISPR sublibrary of guide RNAs targeting host genes important for HIV replication across multiple strains (the HIV dependency factor or HIV-Dep library). The HIV-Dep library has guides targeting 525 genes represented by 8 guides targeting each gene and 210 non-targeting controls (NTCs) [17]. A MetaScape analysis [31] of the HIV-Dep library shows the most enriched gene ontology is chromatin organization, followed by several processes involving gene expression, DNA metabolism, and viral infection pathways (Figure 1A). Genes in many of these categories were previously validated to be important in acute HIV-1 infections [17]. We hypothesized that a subset of these HIV dependency factors is also necessary for the activation of HIV from latency. Thus, to investigate host genes that are required for the reversal of HIV-1 latency, we performed a CRISPR screen using a modification of the HIV-CRISPR system [12,18,19] (Figure 1B). Briefly, this screen in the context of latency reversal relies on transducing latently infected Jurkat T cells (J-Lats) with an HIV-CRISPR lentiviral vector containing a library of sgRNAs. The sgRNAs are flanked by a Ψ-packaging signal, allowing the guides to be packaged into budding virions. We employed this modified latency HIV-CRISPR assay to identify factors important for latency reactivation using two different J-Lat models that contain independently derived integration sites: J-Lat 10.6 and J-Lat 5A8. The goal for this screen was to treat the cells with activating doses of LRAs and deep sequence the supernatant containing the guides compared with the gDNA knockout pool. In contrast to a previous HIV-CRISPR screen where we examined epigenetic factors whose knockout would activate HIV from latency by analyzing guides enriched in the viral supernatant (Figure 1B, scenario 1) [12], in the present screen, the expectation is that genes required for reactivation from latency would be depleted in the viral supernatant relative to the genomic knockout pool (Figure 1B, scenario 2). 

J-Lat cells transduced with the HIV-Dep library were treated with low doses of the non-canonical NF-κB inhibitor AZD5582 (1 nM) and the pan-bromodomain inhibitor I-BET151 (2.5 μM), which led to significant increases in viral production, as measured by reverse transcriptase activity (Figure 1C). Previous studies have also shown that this combination of LRAs is synergistic in the J-Lat model of latency reversal [32]. After deep sequencing the viral supernatant and genomic DNA pool, we used MAGEcK analysis in order to compare the guides enriched or depleted in the supernatant with the genomic knockout pool to identify those genes depleted in the supernatant (Appendix A). We generated a gene set enrichment analysis [20] of our most depleted hits and found the top five enriched pathways in both J-Lat 10.6 and J-Lat 5A8 were related to transcription (Figure 1D). Furthermore, we also saw pathways for RNA splicing and polyadenylation. This is consistent with transcriptional regulation being one of the major axes of host control that underlies the release of HIV-1 from latency. We conclude that our screen can identify and enrich gene pathways that are relevant for the release of the HIV-1 provirus from latency in the presence of AZD5582 and I-BET151 combination treatments.

To understand the role that HIV dependency factors play in terms of latency reactivation, we compared our screens with previous HIV-CRISPR screens that were aimed at identifying factors required for HIV replication in Jurkat cells [17]. A z-score analysis was used as a measure of how depleted genes were in each of the screens and to allow for a cross-comparison regardless of the magnitude of depletion of each guide. Sorting the mean z-score for HIV-1 replication (marked as LAI in Figure 2A) shows that the most depleted genes are *CXCR4* and *CD4,* which are essential for HIV replication but not latency reactivation (Figure 2A, left). This is expected since J-Lat cells are already infected with HIV-1. Other factors that scored highly in the HIV-1 replication screen, but not in the present HIV latency screen, include genes of unknown function in the HIV lifecycle, such as *ATP2A2* and *SS18L2* (Figure 2A, left). In contrast, nearly all of the most depleted factors in the HIV latency screens were also highly depleted in the HIV replication screen (Figure 2A, right, sorted by most depleted in the HIV latency screens; see Appendix A for the complete list of z-scores). We conclude that a subset of HIV dependency factors is required for reactivation from latency.

We chose to validate a subset of the hits in the HIV latency screen that were among the top twenty ranking hits and were shared hits in both J-Lat 10.6 and J-Lat 5A8 cells (Figure 2B, complete list of the screen in Appendix A) by electroporating the Cas9 ribonucleoprotein complex (RNP) complex containing three unique guides against each gene or by lentiviral transduction of single guide RNAs. We tested *CCNT1*, *ELL*, *UBE2M, TBL1XR1, HDAC3*, *AMBRA1, ALYREF,* and *SBDS* gene knockouts (Figure 2C). As a negative control, we included guides targeting the adeno-associated virus integration site 1 (*AAVS1*) “safe harbor” locus, a gene whose disruption does not adversely affect the cell [33], or a non-targeting control (NTC). Pooled knockouts were validated by genomic sequencing and Inference of CRISPR Edits (ICE) analysis (Appendix A). In addition, the CCNT1 pooled knockout was also validated by western blotting (Appendix A).

In the J-Lat 10.6 line, we found that there is reduced reactivation in *CCNT1*, *ELL*, *UBE2M, TBL1XR1, HDAC3*, *AMBRA1,* and *ALYREF* knockouts relative to non-targeting controls and guides targeting a safe harbor locus, *AAVS1* (Figure 2C). We did not see a significant effect in the *SBDS* knockout cells, but interestingly, *AMBRA1* and *ALYREF,* which were less knocked out in pools than *SBDS* in the J-Lat screens, showed a phenotype (Figure 2C). However, the strongest effect on preventing HIV latency reversal was the knockout of *CCNT1,* which was also the top hit in our screen. We conclude that the screen is able to identify genes that are key for latency reactivation in the J-Lat models. 

### 3.2. Cyclin T1 Is Essential for Reactivation from Latency in Both J-Lat and Primary T Cells

Cyclin T1 (CCNT1) is a well-characterized regulator of HIV transcription that binds to the viral protein Tat and TAR [34,35,36] and was the top hit for both J-Lat models (Figure 2). Additionally, CDK9, which binds to Cyclin T1 in the positive transcription elongation factor complex (P-TEFb), is substantially depleted in the CRISPR screen of both cell lines (Appendix A). In order to explore this hit further across a broader range of LRAs, we generated clonal knockout lines of *CCNT1* in the J-Lat 10.6 cell line. The clonal knockouts are completely abrogated of CCNT1 expression, as shown by western blotting and the sequencing of genomic DNA (Figure 3A, left). Moreover, we did not see an upregulation of CCNT2, a paralog of CCNT1 that also binds CDK9 as part of the host P-TEFb complex [14,16] (Figure 3A, right). 

HIV latency is a result of a combination of blocks that prevent transcription initiation and elongation, and LRAs target a broad range of these different facets of proviral gene expression. We explored a range of LRAs in the *CCNT1* clonal knockout lines. We found that CCNT1 is necessary for latency reversal with both CD3/CD28 activation and with Tumor Necrosis Factor Alpha (TNFα) cytokine. Reactivating with CD3/CD28 and TNFα are mechanisms that result in the upregulation of NF-κB signaling, emphasizing the transcriptional initiation component of latency. Therefore, we explored additional means of reactivation including AZD5582 and I-BET151 together, Prostratin—an activator of PKC and known inducer of P-TEFb activity [37,38]—and SAHA/Vorinostat [39], the histone deacetylase inhibitor (HDACi) (Figure 3B). In all treatments, wild type cells for CCNT1 were able to reactivate, but *CCNT1* knockout prevented latency reactivation with each LRA. We conclude that CCNT1 is essential for reactivation from latency for multiple diverse mechanisms of latency reversal in J-Lat cells.

We also investigated the role of CCNT1 in latency reactivation in primary CD4+ T cell lymphocytes isolated from healthy donors. We first activated and infected peripheral blood CD4+ T cell lymphocytes with an HIV-1 dual-reporter virus previously described [12]; the first marker is a destabilized GFP reporter, which is a marker of active provirus expression. The destabilized GFP has a short half-life and thus is indicative of active expression of the provirus. The second marker, the Thy1.2 (mouse CD90) viral reporter, is a cell surface marker that allows us to determine cells that have, at one point, been infected. This cell surface marker has a slow turnover and persists over the latency establishment period, and thus marks cells that have been infected with the dual reporter virus but may not be actively producing the virus. After infection with a dual-reporter virus, the infected cells were knocked out by electroporation with Cas9 and gRNA for *CCNT1* or control *AAVS1*. The cells were cultured for an additional two weeks to enter latency and then measured for the capability for latency reactivation after LRA treatment as determined by flow cytometry for dual positive GFP and CD90 expression (Figure 3C). 

We tested knockouts from three independent donors with the potent LRA combination phorbol 12-myristate 13-acetate (PMA) and ionomycin, as well as with CD3/CD28 antibody co-stimulation (Figure 3C for all donors, Appendix A for the gating of one donor as an example). In the control *AAVS1* knockout, we found that there is an increase in the percentage of total cells that are both Thy1.2+ and GFP+ when treated with PMAi or CD3/CD28 co-stimulation, indicating an increase in cells that have active transcription of viral genes (5.46% without LRA, 39.7% with LRA) (Figure 3C). In contrast, the *CCNT1* knockouts had a stark reduction in Thy1.2+ and GFP+ cells when treated with PMAi and CD3/CD28 co-stimulation relative to *AAVS1* knockout (Figure 3C, Appendix A). We also noted that there is a modest reduction in Thy 1.2+ GFP+ cells in the *CCNT1* knockout that have not been treated with PMAi or CD3/CD28 co-stimulation. This is consistent with our previous result in clonal knockouts in J-Lat cells, suggesting that minimal levels of HIV-1 transcription that occur in latent cell populations are lower in *CCNT1* knockouts. We conclude that Cyclin T1 is an essential gene for latency reactivation. 

To exclude the possibility that Cyclin T1 blocks the ability for CD4+ T cells to activate, as well as ensure T cell activation is occurring properly in our experiments, we simultaneously stained cells for the early activation marker CD69. PMAi and CD3/CD28 co-stimulation both show a significant degree of activation over unstimulated cells. We saw no significant change between *AAVS1* and *CCNT1* knockout in any of the conditions (Figure 3D). We conclude that *CCNT1* is key for latency reactivation in primary CD+4 T cells but does not affect the ability of these cells to be activated upon stimulation.

### 3.3. Cyclin T1 Is Non-Essential in T Cells and Regulates Host Genes to a Much Lesser Extent than It Regulates HIV-1

Given that P-TEFb has been reported to be required for transcription elongation of many host genes [40], we were initially surprised that the knockout of *CCNT1* is viable. However, we did not see a drastic change in cell growth measured over a span of nine days (Figure 4A). This led us to broadly investigate the role of Cyclin T1 in transcription in T cells by performing bulk RNA sequencing of J-Lat 10.6 cells and two independent clonal knockouts of *CCNT1* in the J-Lat 10.6 cells either without an LRA or treated with TNFα. As a control, we first compared the RNA sequencing data from wild type J-Lat 10.6 line that has been treated with TNFα versus the J-Lat 10.6 line (*CCNT1* is wild type in both cases). HIV-1 transcripts are among the most significantly upregulated genes in the TNFα treatment for wild type *CCNT1* in J-Lat 10.6 cells (Figure 4B). We also see an upregulation of *PGLYRP4, RELB,* and *BCL3*, which are genes related to NF-κB signaling or otherwise known to be upregulated by TNFα (Figure 4B) [41,42,43]. We next examined how HIV-1 and host gene transcripts are affected in TNFα-treated cells that have *CCNT1* knocked out relative to TNFα-treated J-Lat 10.6 cells that are wild type for *CCNT1* (Figure 4C). Strikingly, RNA transcripts related to HIV-1 genes in *CCNT1* knockout are the most depleted transcripts over any host gene, relative to wild type *CCNT1* (Log_2_(FC) = −10.92) (Figure 4C). Even in the absence of LRA, we find that HIV-1 transcripts are the most depleted relative to other host genes (Log_2_(FC) = −9.29) when comparing *CCNT1* knockout versus wild type (Figure 4D). Thus, the basal transcription of HIV-1 transcripts that occur in J-Lat lines is highly dependent on Cyclin T1. Regardless of TNFα treatment, the host genes that were highly depleted in *CCNT1* knockout included *FAM222A-AS*, *GGTLC1*, *MYO10, NETO1*, and *ZBTB16*. Notably, we did not find significant upregulation of *CCNT2* transcripts in the *CCNT1* knockout versus wild type (Log_2_(FC) = 0.078) or the LRA-treated cells (Log_2_(FC) = 0.139). Nonetheless, *CCNT1* knockout affects the HIV-1 provirus far more than any other transcriptional unit in the J-Lat cells.

We further investigated the effect of *CCNT1* knockout on uninfected primary CD4+ T cells. *CCNT1* was knocked out by the electroporation of *CCNT1* guides complexed with Cas9 in three independent donors, and the knockout was validated to be over 90% by sequence analysis (Appendix A). The *AAVS1* locus was knocked out in parallel as a control. Similar to the primary cell latency model (Figure 3C), we found that the *CCNT1* knockout did not affect the expression of the CD69 activation marker after treatment with anti-CD3/anti-CD28 beads (Figure 5A). As expected, a comparison of RNA sequencing on the primary cells stimulated with anti-CD3 and anti-CD28 antibodies versus the unstimulated cells shows dramatic upregulation and downregulation of genes (Figure 5B); for example, there is an upregulation of IL31, which is a cytokine known to be upregulated by activated T cells [44]. However, the same RNA-seq analysis of the *AAVS1* knockout cells compared to the *CCNT1* knockout cells upon stimulation with anti-CD3/anti-CD28 beads shows that the *CCNT1* knockout cells have the same expression profile as the control knockout cells, i.e., there are no significant differences in upregulated or downregulated genes in the comparison (Figure 5C) when *CCNT1* is knocked out. We also compared RNA expression profiles of the *CCNT1* knockout cells with the control *AAVS1* knockout cells in the absence of anti-CD3 and anti-CD28 stimulation, and again found very few genes that are upregulated or downregulated (Figure 5D). In addition, the magnitude of these gene expression changes was minimal. As an example, the most enriched gene for *CCNT1* knockout compared to *AAVS1* knockout has a −log_2_FC less than 2, and the most depleted gene has a −log_2_FC greater than −2 (Figure 5D). Thus, we conclude that there are minimal changes in gene expression when *CCNT1* is knocked out in primary CD4+ T cells with and without T cell receptor stimulation. Together, we conclude that *CCNT1* does not play an essential role in peripheral primary CD4+ T cells.

## 4. Discussion

We used an HIV-CRISPR screening approach to identify host genes required for the activation of HIV from latency starting from the hypothesis that a subset of host genes previously identified as being necessary for HIV replication are also necessary for HIV reactivation from latency. Among the genes identified are many genes involved in transcription elongation, transcription initiation, and protein degradation. The top hit in our screens was Cyclin T1 (*CCNT1*), which we show is essential for reactivation from latency across a wide range of latency reversal agents of different mechanisms of action, as well as in primary T cells. In contrast, *CCNT1* appears to be redundant with other host genes for normal transcriptional regulation in T cells and is, therefore, an attractive target for specifically silencing integrated HIV-1 proviruses.

### 4.1. Cyclin T1 Is Much More Important for HIV Latency Reversal than T Cell Biology In Vitro

Despite the described role of Cyclin T1 and the P-TEFb complex in host gene transcription, we were able to generate knockout clones of *CCNT1* without affecting cell growth and viability. We also did not see a significant upregulation of CCNT2 protein expression. Collectively, we interpret our results to mean that *CCNT1* is dispensable in T cells and that *CCNT2* or *CCNK* may compensate for the loss of *CCNT1*. One model is that there are redundant mechanisms that govern the transcription elongation of host genes. Previous work on *CCNT1* and *CCNT2* knockouts in mice illustrated unique phenotypes, initially suggesting the possibility that these two genes have separate functions despite both being able to form the P-TEFb complex [45,46]. RNA sequencing of CCNT1 and CCNT2 knockdowns by another group using shRNA in HeLa cells also suggested that these two proteins regulate different sets of genes [47]. However, while CCNT1 had very large effects on HIV-1 transcripts, we found that CCNT1 has minimal effects on host gene transcription in Jurkat T cells. We observed a modest downregulation of several host genes including *GGTLC1*, *MYO10*, *NETO1*, *ZBTB16*, and *BZRAP1*. GGTLC1 is a metabolic enzyme and member of the gamma-glutamyl transpeptidase family, of which there are several paralogs [48]. Myo10 is an unconventional myosin that is associated with actin and filopodia. This gene has ubiquitous but low expression across tissues [49] but has been reported to promote HIV-1 infection in human monocyte-derived macrophages [50]. *ZBTB16* (also known as *PLZF*) is a transcription factor and is known to be important for natural killer T cells but is repressed in non-innate T cells and not upregulated in T cell activation [51]. Collectively, we see slight changes in gene expression in J-Lat cells on *CCNT1* knockout that led to drastic changes in HIV-1 gene expression, but few host genes seem to be affected by knockout.

On the other hand, there were no significant changes in gene expression of *CCNT1* knockout versus *AAVS1* knockout in primary CD4+ T cells activated with CD3/CD28 co-stimulation. Knockouts of *CCNT1* in primary CD4+ T cells also had little effect on cell viability and the cell surface expression of an activation marker, CD69. In the unstimulated condition, we see some low-magnitude gene expression changes. *AIF1L* is a mildly downregulated gene, and to date, there is no clear known function of this gene in T cell biology. In human podocytes, this gene is known to function in actomyosin contractility, and thus cells that lack this gene have increased filopodia [52]. Upregulated genes include *IL5*, *DMD*, *STRA6*, *ENOX1*, and *DEPP1*. None of these genes are particularly implicated in T cell biology. Mutations in the *DMD* (Dystrophin) gene are implicated in Duchenne’s Muscular Dystrophy, an X-linked recessive disorder. We also saw the upregulation of *MYOF* (Myoferlin), a gene whose mutations are associated with muscle weakness [53,54]. An interesting possibility is that *CCNT1* positively and negatively regulates genes associated with muscle function, given that we saw an upregulation of genes implicated in muscle disease and a downregulation of *MYO10* in the J-Lat 10.6 RNA sequencing data on *CCNT1* knockout.

We reason that while CCNT1 and CCNT2 gene regulation may have tissue-specific contexts, CCNT1 is likely non-essential in CD4+ T cells. Data from DepMap indicate that *CCNT1* is classified as “strongly selective”, indicating there are cell lines in which this gene is more essential, but *CCNK* is considered widely essential in most CRISPR screens [53]. Previous work suggests that CCNT1 is targeted by proteasomal degradation in resting CD4+ T cells, and thus CCNT1 protein expression in resting CD4+ T cells is low [54,55,56], but our data suggest that it is not necessary for T cell activation. While we saw little effect of *CCNT1* knockout on host RNA transcripts in a relevant target cell type for HIV-1 infection, we cannot rule out the possibility that *CCNT1* plays a key role in host biology in more differentiated T cell functions or other HIV-1-prone cell types, including macrophages and glial cells. We interpret this to mean that the role of *CCNT1* may be redundant in T cells for host gene expression but not HIV-1 activation. 

CCNT1 binds to the viral Tat protein, which subsequently binds to the TAR region and enables the transcription initiation of proviral genes. Previous work suggests that CCNT2 can bind to the viral Tat protein, but this complex does not bind to TAR and thus cannot initiate viral transcription. The mutation of a single amino acid residue asparagine 260 to cysteine in CCNT2 was sufficient to rescue the function of Viral Tat and TAR interaction [16]. Similarly, the mutation of a tyrosine residue at 261 to cysteine in mouse Cyclin T1 protein rescues Tat/TAR function [57,58]. Thus, the level of specificity is not at the Tat interface but rather at the TAR interface, which might be an evolutionary advantage for the virus to resist host escape. 

### 4.2. Other Hits in the HIV-CRISPR Screen

Several genes involved in transcription are among our most depleted genes. Notably, NFKB1—the transcription factor that binds to 5′ LTR to allow for the transcription initiation of proviral genes, is among our top hits. We also see other transcription-related genes depleted in both cell lines. ELL—an elongation factor for RNA polymerase II and a component of the super elongation complex—is the second most-depleted hit. We also note that there are several post-translational modifying enzymes that are novel in terms of latency reactivation. The Ubiquitin Conjugating Enzyme E2 M (UBE2M) is highly depleted and is known to be involved in the neddylation pathway, which uses a ubiquitin-like conjugation process. UBA3, which makes up the E1 enzyme of the neddylation conjugation pathway, also is depleted but to a lesser degree. Both of these neddylation genes were also depleted in our previous CRISPR screen on Jurkat T cells to identify dependency factors, and *UBE2M* was validated for several strains of HIV [17]. Histone Deacetylase 3 (HDAC3) forms a complex with TBL1XR1 as part of the SMRT N-CoR (nuclear coreceptor complex), which regulates the modification of histones and gene regulation [59,60,61]. siRNA studies of TBL1XR1 have found redundancy with its paralog TBL1X, whereas HDAC3 was found to be essential. Vorinostat, a commonly used LRA, targets HDAC3 along with Class I and Class II HDACs [62]. It is unclear why *HDAC3* knockout may prevent latency reactivation, but we reason that latency reactivation depends in part on a noncatalytic activity of *HDAC3*.

A genome-wide CRISPR screen was previously performed that identified factors important for latency reversal [63]. In that study, the authors generated a pool of latently infected cells and performed a whole genome CRISPR knockout screen, treated with a panel of different LRAs, sorted the GFP cells, and identified genes specific for latency reversal as well as common genes required regardless of reactivation approach. In comparing our screens, we find that many of our hits are shared with the “common” cluster of genes where they tested TCR cross-linking, TNF-a, PMAi, and AZD5582 as LRAs and identified the common genes required for reactivation: *CCNT1, HDAC3, NFKB1, MBNL1, UBE2M, TBL1XR1, UBA3, AMBRA1, SBDS,* and *MED7*. Thus, despite only screening with AZD5582 and I-BET151, we are able to identify several hits that promote latency reactivation regardless of the LRA used. *UBA3* and *UBE2M* are of interest, as they are both components of the neddylation pathway [64], and while *NEDD8* is not in our HIV-DEP gene library, the whole genome screen identified *NEDD8* as a hit in its AZD5582 screen [63]. In contrast, there are several hits that are depleted and validated in our more targeted screens but not the whole genome screen, such as *ELL* and *ALYREF* (Figure 2). Nonetheless, there is good agreement between screens, validating the approach of searching for host factors involved in latency through CRISPR screens combined with LRAs. 

### 4.3. HIV Dependency Factors Versus Host Genes Necessary for Latency Reversal

Our initial hypothesis was that HIV-1 dependency factors may play a role in latency reactivation given the importance of transcription in establishing infection and that transcription is a major facet that contributes to latency. Consistent with our hypothesis, we find that a large proportion of genes are important as both HIV dependency factors and HIV latency reversal factors (Figure 2A). While transcription is the major category of genes in our screens (Figure 1C,D), the factors span beyond transcription; we find factors involved that are key for reactivation, including UBA3, UBE2M, AMBRA1, and ALYREF. In contrast, we also observe factors that are important as HIV-1 dependency factors but not in latency reactivation, including ATP2A2, SS18L2, SMARCB1, and PCGF1, whose guides were depleted in Jurkat T cell screens but not J-Lat screens. ATP2A2 is a calcium transporting ATPase that was found to be upregulated during the G1/S phase of the cycle by Tat, but its role in the viral life cycle is otherwise unknown [65]. Similarly, *SS18L2* mRNA was found to be upregulated in HIV-1 in early infection, as was found from the RNA profiling of CD4+ and CD8+ T cells in people living with HIV-1 versus those who were either nonprogressors or control HIV-1 negative groups [66]. SMARCB1 is a component of the SWI/SNF chromatin remodeling complex, along with INI1 (Integrase Interactor-1), and is known to play many roles in HIV-1 replication, including integration, transcription, and particle maturation [67]. *PCGF1* (polycomb group RING finger protein 1) was also depleted in HIV-1 dependency factor screens, but not in J-Lat screens in this study. Polycomb group proteins largely lead to transcriptional repression through the methylation of histones, and thus are thought to contribute to HIV-1 latency. This might contribute to the opposite phenotype we see in this study versus infection screens; PCGF1 may play a role in maintaining latency but is required for establishing infection. An interesting possibility is that PCGF1 is required for infection as it helps to establish a chromatin landscape that leads to either productive transcription at the integrated provirus or even transcriptional silencing, which may ultimately contribute to HIV-1 latency. Collectively, the latency HIV-CRISPR screens can help to narrow down the stage of the viral life cycle where dependency factors play a role, and give insight into novel latency reversal factors.

### 4.4. Gene Paralogs in a “Block and Lock” Latency Approach

Our latency HIV-CRISPR screen in this study revealed our top hit, *CCNT1,* was able to be knocked out with little effect on T cell biology, which was likely due in part to its paralogs *CCNT2* and *CCNK*. Cyclin T1 and T2 are paralogs that have sequence similarities at the amino acid level in the N-terminal region (81% identity), but the C-terminal domain is more divergent and far less similar (~46% amino acid identity between CCNT1 and CCNT2 [14,15]. Cyclin K also is similar in the N-terminal domain but has a shorter C-terminal region and therefore a much smaller protein. One possibility is that there is a sequence in this C-terminal domain that adds specificity for gene regulation in CCNT2 or CCNK that allows for the regulation of host genes and recruitment to different cellular promoters. Collectively, our findings suggest that CCNT1′s paralogs are sufficient for the transcription elongation of host genes, but CCNT1 is required for the transcription of HIV-1 genes.

This approach to “block and lock,” whereby a factor is required for viral replication but not for host function, may be a path forward in further identifying gene targets to inhibit HIV-1 viral reactivation. Separate but parallel approaches have been used in cancer contexts, whereby synthetic lethality is exploited to promote the death of cancer cells. A recent study has led to the identification of paralogs with redundant functions that lead to cell death when a pair of gene paralogs are knocked out [68]. In this study, 12% of paralogs tested led to cell death in their context. We interpret this to mean that there are a great number of gene paralogs that may serve redundant functions. Ongoing work will seek to identify factors that are like *CCNT1* in that when targeted, have drastic effects on viral replication and minimal effects on the host by focusing on the top hits that have gene paralogs and thus may have redundancy. Other screen hits had gene effect scores similar to *CCNT1* on the DepMap Portal [53], including *TBL1XR1*, *OTUD5* and *AMBRA1*, suggesting that these may either have paralogs or dispensable functions for cell biology.

While LPAs have been developed in a block and lock approach, this approach still remains a challenge. In the case of dCA, for instance, HIV confers resistance to this drug through mutations in the LTR, Nef, and Vpr [69,70]. Targeting CCNT1—or additional gene paralogs with redundant functions—may prove to be a strong complement to these LPAs, given how drastic an effect that *CCNT1* knockouts have on HIV-1 replication. Although the shock and kill approach and the discovery of LRAs have been a large area of focus in recent years, there may be a role for both approaches in permanently silencing the latent reservoirs in those tissue reservoirs that are resistant to LRAs. Further investigation of *CCNT1* knockout in macrophages, microglial cells, and other resident tissues, as well as other genes that have redundancy in a similar regard as *CCNT1*, will provide a good path forward to identify additional block and lock mechanisms that may supplement other approaches to an HIV functional cure.

## Figures and Tables

**Figure 1 viruses-15-01863-f001:**
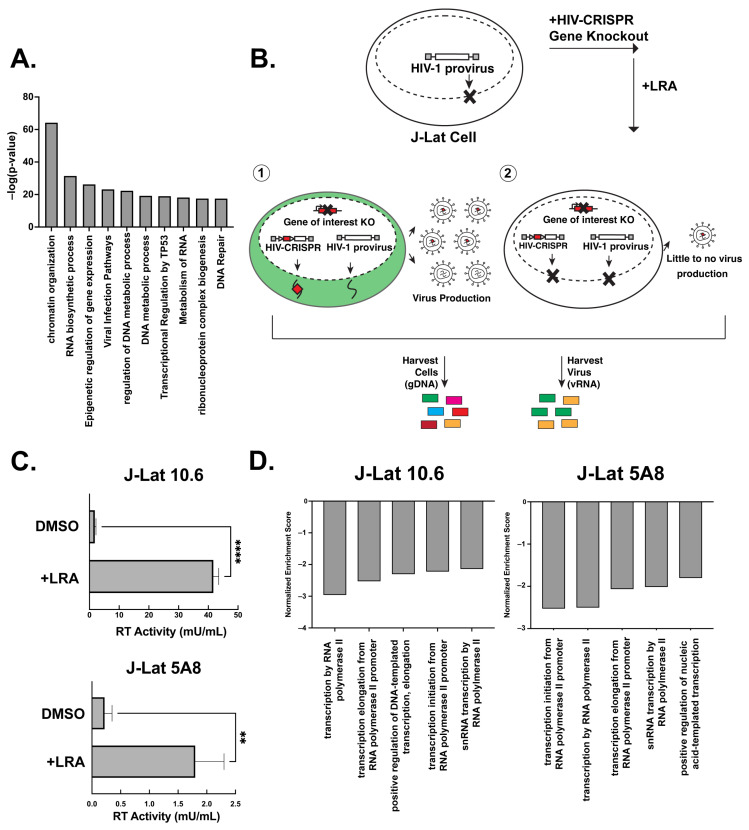
A latency HIV-CRISPR screen to identify the factors required for latency reversal. (**A**) A MetaScape analysis of the genes in the HIV-Dep gene library is shown, with enriched pathways on the x-axis and statistical significance on the y-axis. (**B**) Overview of the latency HIV-CRISPR screen of HIV dependency factors. The HIV-CRISPR vector has intact 5′ and 3′ LTRs and can be packaged by HIV-1 after integration [19]. J-Lat cells were transduced with an HIV-CRISPR library of genes of HIV-1 dependency factors, selected for integration by puromycin selection, and treated with a latency reversal agent (LRA). Viral RNA (vRNA) and genomic DNA (gDNA) are harvested at the end of the experiment. Guides corresponding with genes that do not affect reactivation from latency are packaged in virions and enriched in the supernatant relative to the genomic DNA pool (scenario 1, left). For genes that are important for latency reactivation after treatment of cells with an LRA, these guides will be depleted in the viral supernatant relative to the genomic DNA knockout library (scenario 2, right). (**C**) Supernatants from J-Lat cells transduced with the HIV-DEP gene library were measured for reverse transcriptase (RT) activity after treatment with the LRA combination AZD5582 (1 nM) and I-BET151 (2.5 μM). Error bars represent technical triplicates, and an unpaired t-test was used for statistical analysis. *p*-value < 0.01 = **, < 0.0001 = ****. (**D**) MAGEcKFlute [20] was used to analyze the screen results of the depleted genes. The normalized enrichment score is on the y-axis (negative because guides to these genes are depleted from the viral supernatant), and the x-axis is the biological processes.

**Figure 2 viruses-15-01863-f002:**
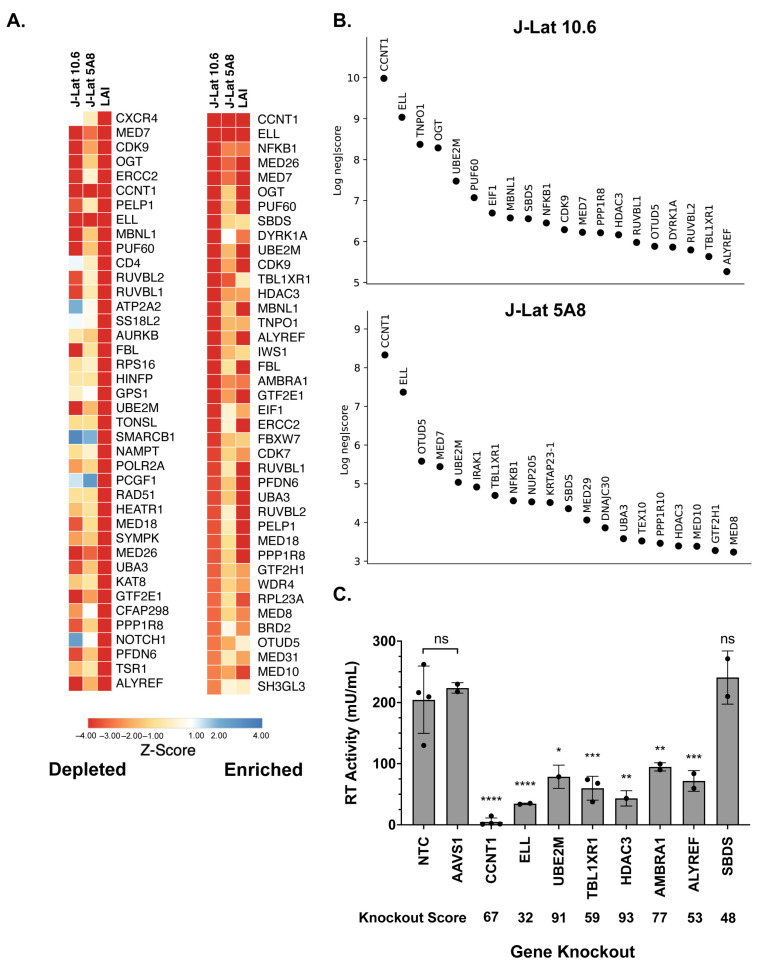
Analysis and validation of the top hits from the HIV-CRISPR screen. (**A**) Z-score analysis of the depleted versus enriched guides across multiple screens. J-Lat 10.6 and J-Lat 5A8 are screens from this study, whereas LAI represents Jurkat cells infected with an LAI strain of HIV-1 from a previous screen performed using the same gene library in Jurkat cells to identify HIV dependency factors [17]. Z-scores are sorted by the most depleted genes in the LAI screen (left panel) and by the most depleted genes in the J-Lat 10.6 line from this study (right panel). The mean z-score of the two replicates, J-Lat 10.6 and J-Lat 5A8, and four replicates of the LAI screen is shown. The most depleted genes are red and the most enriched genes are blue. Z-scores that were less than −4 were capped at −4 in the heatmap. (**B**) The top 20 most depleted hits from each J-Lat line in ranked order are shown. (**C**) Selected hits from the screen were tested by performing gene knockouts (x-axis), treating with the LRA combination AZD5582/I-BET151, and assaying for reverse transcriptase activity. Gene knockouts were performed using a lentiviral knockout approach and/or an electroporation with Cas9 and RNPs. Each point represents a single lentiviral or electroporation knockout experiment conducted in triplicate. The average of RT activity from two guides targeting each gene was taken for lentiviral knockouts, and the electroporation knockouts included three individual guides targeting each gene. The ICE gene knockout score for each experiment was averaged and is shown below each gene on the x-axis. Statistical analysis was performed using a two-way ANOVA and Šídák’s multiple comparisons test to measure the difference in latency reactivation between each gene knockout relative to the NTC and AAVS1 controls combined. *p*-value ≥ 0.05 = ns (not significant), <0.05 = *, <0.01 = **, <0.001 = ***, <0.0001 = ****. NTC/*AAVS1* controls are combined; each dot represents either an *AAVS1* or NTC control for an individual experiment. Each experiment (dot) has three technical replicates: NTC, n = four experiments, three replicates each; *AAVS1*, n = two experiments, three replicates each; *CCNT1*, n = four experiments, three replicates each; *ELL*, n = two experiments, three replicates each; *UBE2M*, n = one experiment, three replicates each; TBL1XR1, n = three experiments, thee replicates each; *HDAC3*, one experiment, three replicates each; *AMBRA1* n = two experiments, three replicates each; *ALYREF*, two experiments, three replicates each; *SBDS* n = two experiments, three replicates each.

**Figure 3 viruses-15-01863-f003:**
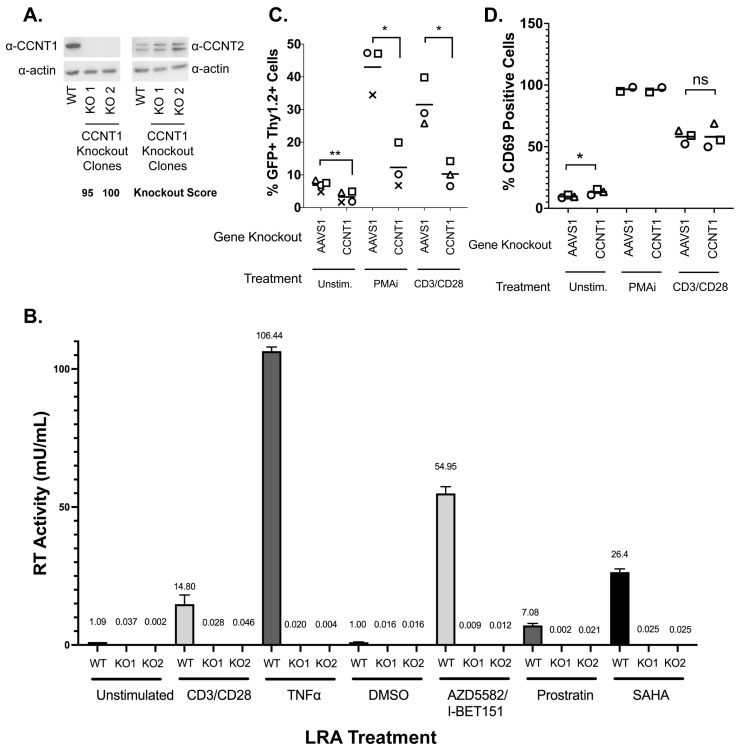
*CCNT1* is required for the reactivation of HIV-1 from latency in Jurkat T cells and primary CD4+ T cells from healthy donors. (**A**) Western blot of cell lysates of J-Lat 10.6 wild type or clonally knocked out for *CCNT1* is shown, with two separate knockout clones. Actin was used as the loading control. Left: the CCNT1 antibody is shown; right: the CCNT2 antibody is shown. ICE knockout scores are shown for each knockout clone of *CCNT1.* (**B**) J-Lat 10.6 cells wild type for CCNT1 and the two clones knocked out for *CCNT1* were treated with the LRAs, as shown on the bottom. The mean of RT activity in the supernatant 24 h after LRA treatment is shown on the y-axis and above each bar. The averages and standard deviation of the experiment conducted in triplicate are represented. (**C**) Primary CD4+ T cells from three different healthy donors were infected with a dual-reporter virus that monitors cells’ active and latent infection. Cells were either knocked out for the *AAVS1* control or *CCNT1* and were either untreated, stimulated with PMAi, or stimulated with anti-CD3/anti-CD28 antibodies at the end of latency establishment. Each shape represents an individual donor. (**D**) CD69 expression was monitored with the different LRA treatments. Each shape represents an individual donor. *CCNT1* ICE knockout scores were 80, 76, 53, and 37 for each of the four donors for CD3/CD28 and two donors for PMAi. A paired t-test was used for the comparison of *AAVS1* knockout vs. *CCNT1* knockout between donors. *p*-value ≥ 0.05 = ns, <0.05 = *, <0.01 = **.

**Figure 4 viruses-15-01863-f004:**
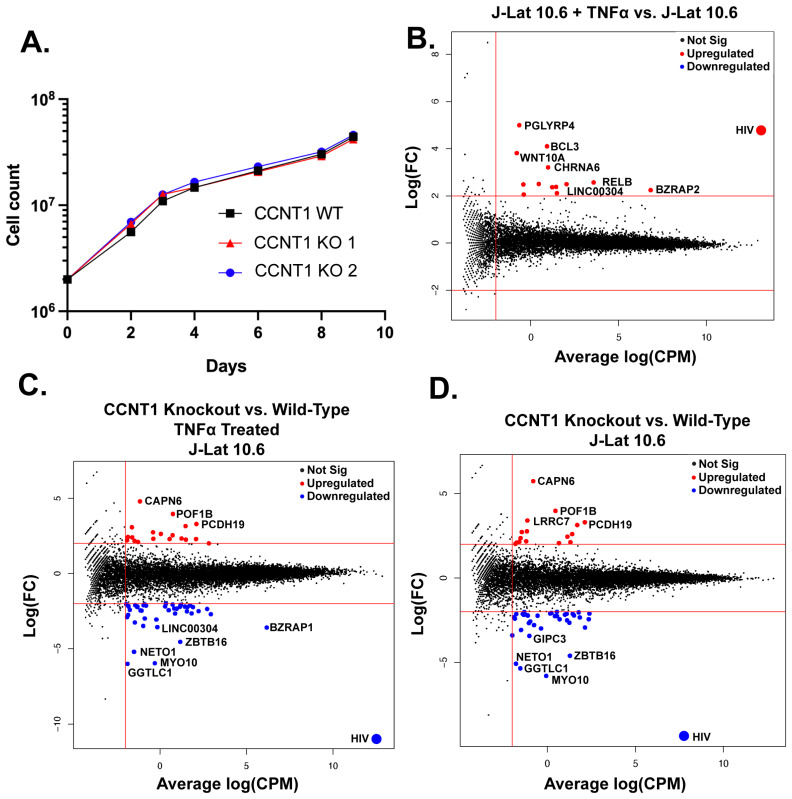
Cell proliferation and RNA sequencing analysis of *CCNT1* knockouts in J-Lat 10.6 cells. (**A**) Cell counts were monitored over a span of nine days in J-Lat 10.6 cells in WT or clonally knocked out *CCNT1* cells. The average of three experimental replicates is shown with standard deviation. (**B**–**D**) Log_2_ FC (fold change) is plotted on the y-axis with the average Log_2_ CPM (counts per million) across technical replicates on the x-axis. Red lines signify genes that have an average Log_2_ CPM > −1, and a |Log_2_ FC| > 2. Red dots signify upregulated genes, whereas blue genes signify downregulated genes for each comparison. (**B**) Differential gene expression of J-Lat 10.6 with TNFα treatment versus J-Lat 10.6 (untreated) is shown. (**C**) J-Lat 10.6 *CCNT1* KO cells (two independent clones each tested in technical triplicate and averaged) versus the J-Lat 10.6 wild type cells—both were treated with the LRA TNFα, and gene expression comparison is shown. (**D**) J-Lat 10.6 *CCNT1* KO cells versus wild type *CCNT1* differential gene expression are shown—neither cell line was treated with an LRA.

**Figure 5 viruses-15-01863-f005:**
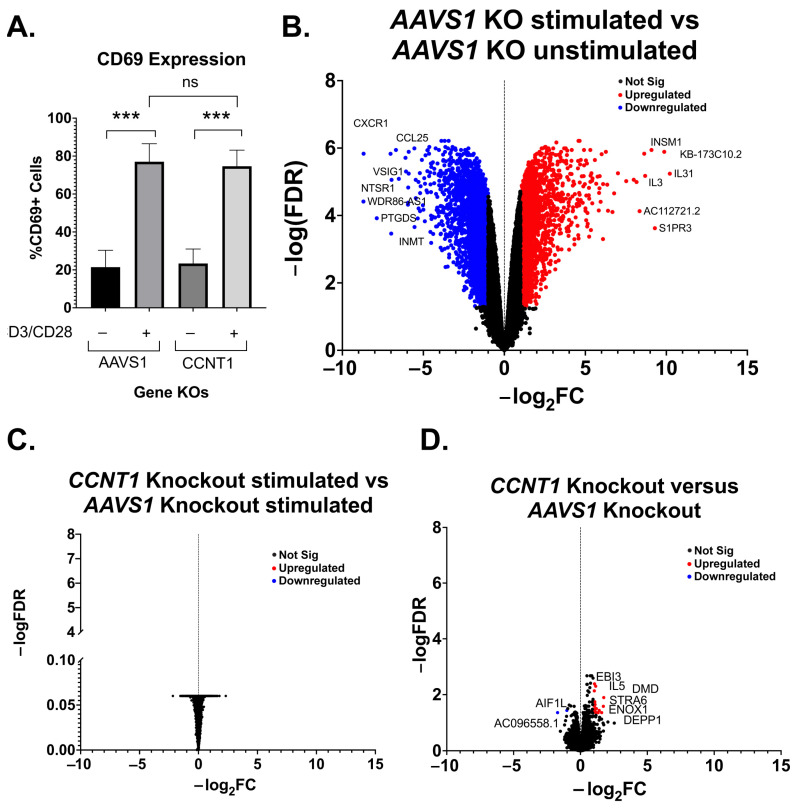
Primary T cell transcripts are largely unaffected by *CCNT1* knockout. (**A**) Uninfected CD4+ T cells from three donors were knocked out for *AAVS1* or *CCNT1* and then treated with CD3/CD28 antibody co-stimulation. Cells were analyzed by flow cytometry to measure CD69 expression. One-way ANOVA was used for analysis with Dunnett’s multiple comparison tests. (**B**–**D**) Volcano plots of primary CD4+ T cell RNA sequencing data are shown, with −log_2_FC shown on the x-axis and −log(FDR) on the y-axis. RNA was isolated from three biological replicates. An FDR = 0.05 was used as a cutoff for significance, and the cutoff for significant gene expression was |Fold-Change| > 1. A subset of genes for each condition are marked that have significance. (**B**) Differential gene expression between *AAVS1* knockout stimulated with CD3/CD28 versus unstimulated is shown. (**C**) A comparison of *CCNT1* versus *AAVS1* knockout is shown, and both were stimulated with anti-CD3/anti-CD28 antibodies. (**D**) *CCNT1* versus *AAVS1* knockout is shown, and neither of these are stimulated with anti-CD3/anti-CD28 antibodies. *p*-value ≥ 0.05 = ns (not significant), <0.001 = ***.

## Data Availability

Next-generation sequencing data are available on the NCBI Gene Expression Omnibus Database (GEO) under the SuperSeries accession number GSE240899. CRISPR screen data are under accession number GSE240894 and RNA sequencing data are under accession number GSE240896.

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
