# Peer review of "A CRISPR Screen of HIV Dependency Factors Reveals That CCNT1 Is Non-Essential in T Cells but Required for HIV-1 Reactivation from Latency"

_viruses, 2023, doi:10.3390/v15091863_

Round 1
Reviewer 1 Report
The paper by Hafer et al. utilizes a CRISPR screen described previously by the lab in the context of JLAT cells to identify factors that are necessary for HIV latency reversal. With this approach, they identify several genes that inhibit reactivation of HIV provirus in cell lines and a primary model following treatment with latency reversing agents. The gene they focus on and validate is CCNT1, which encodes for Cyclin T1, a subunit of P-TEFb, The importance of cyclin T1 and P-TEFb in Tat-mediate HIV transcription has been well characterized, and intuitively it is not surprising this is a prominent hit in their screen. What is intriguing is the specificity of CycT1 for latency reversal and that knocking it down had minimal impact on overall T cell expression, which, as they speculate, maybe reflects the redundant functions of Cyclins. Overall, this is a clearly written paper and, although the CCNT1 observation may not be novel and is predictable, the overall results are thought provoking especially regarding the potential cell-specific activities of different P-TEFb complexes. Below are a few questions and suggestions for the authors to consider.
11. It seems as if it would be appropriate to discuss in greater detail the specificity of Tat interactions with P-TEFb through CyclinT1. I wonder if the authors have speculated why Tat would evolve to preferentially interact with this cyclin?
22. Experiments looking at whether deletion of CCNT1 altered RNAP II and nucleosome positioning on the LTR, could provide insights into the role of P-TEFb in coordinating these transcriptional events and whether they are coupled or can be uncoupled. Such experiments would also assure that CCNT1 was acting through the predicted mechanism of transcriptional elongation.
33. Were CCNT2 and/or CCNK knocked down and did they have an impact on T cell gene expression?
44. Using the primary model, were there differences observed between controls and deleted CCNT1 cells in the time of latency establishment?
55. Very minor, I did not see a description for supplemental figure 1.
Author Response
- It seems as if it would be appropriate to discuss in greater detail the specificity of Tat interactions with P-TEFb through CyclinT1. I wonder if the authors have speculated why Tat would evolve to preferentially interact with this cyclin?
We thank the reviewer for this suggestion. We now added more information in the discussion about the specificity of CyclinT 1 and T2 for Tat versus the TAR structure as well as some speculation about the evolutionary advantage for the virus. Our changes in the manuscript are in bold:
“...We interpret this to mean that the role of CCNT1 may be redundant in T cells for host gene expression but not for HIV-1 activation. CCNT1 binds to the viral Tat protein which subsequently binds to the TAR region and enables transcription initiation of proviral genes. Previous work suggests that CCNT2 can bind to the viral Tat protein, but this complex does not bind to TAR and thus cannot initiate viral transcription. Mutation of a single amino acid residue asparagine 260 to cysteine in CCNT2 was sufficient to rescue function of Viral Tat and TAR interaction (Bieniasz and Cullen et al. Journal of Virology 1999). Similarly, mutation of a tyrosine residue at 261 to cysteine in mouse Cyclin T1 protein rescue Tat /TAR function (Bieniasz and Cullen et al. EMBO J 1998, Garber and Jones et al. Genes Dev 1998). Thus, the level of specificity is not at the Tat interface, but rather the TAR interface which might be an evolutionary advantage for the virus to resist host escape.”
- Experiments looking at whether deletion of CCNT1 altered RNAP II and nucleosome positioning on the LTR, could provide insights into the role of P-TEFb in coordinating these transcriptional events and whether they are coupled or can be uncoupled. Such experiments would also assure that CCNT1 was acting through the predicted mechanism of transcriptional elongation.
We agree that these are very interesting extensions of our work. However, we feel that the present system of knockout is not the most appropriate way to address the questions of how CCNT1 affects RNAP II positioning because it gives no information on the timing of the changes of nucleosome positioning. We would prefer to examine this question using either a degron system where CCNT1 can be eliminated and RNAP II can be examined as a function of loss, and/or a knockout combined with an inducible promoter for CCNT1 production so that positioning can be examined as a function of gain of CCNT1. However, both of these approaches are beyond the scope of the present manuscript and will take time to set up and analyze in a more extensive study.
- Were CCNT2 and/or CCNK knocked down and did they have an impact on T cell gene expression?
We did not do this experiment because our focus was on HIV latency control and neither CCNT2 nor CCNK scored highly on our screen for HIV Dependency Factors (Montoya and Emerman et al. mBio 2023).
- Using the primary model, were there differences observed between controls and deleted CCNT1 cells in the time of latency establishment?
We have not looked explicitly at the latency establishment period to test whether CCNT1 knockout may promote latency establishment. However, we now point out in our manuscript that there are fewer Thy 1.2+ GFP+ cells in the CCNT1 knockout cells compared to the control knockout cells even in the absence of LRA treatment, which suggests CCNT1 may also play a role in latency establishment. See text below for added language.
“In contrast, the CCNT1 knockouts had a stark reduction in Thy1.2+ and GFP+ cells on treatment with PMAi and CD3/CD28 co-stimulation relative to AAVS1 knockout (Figure 3C, S1). We also noted that there is a modest reduction of Thy 1.2+ GFP+ cells in the CCNT1 knockout that have not been treated with PMAi or CD3/CD28 co-stimulation. This is consistent with our previous result in clonal knockouts in J-Lat cells suggesting that minimal levels of HIV-1 transcription that occur in latent cell populations are lower in CCNT1 knockouts.”
- Very minor, I did not see a description for supplemental figure 1.
We are not sure why the figure legend for the supplemental figure was not included in the compiled manuscript. We have checked to make sure that it is now included
Reviewer 2 Report
This manuscript, Hafer et al, describes results from a CRISPR knockout screen to identify genes required for reactivation of HIV-1 provirus by various latency reversing agents. The screen involved use of a subset CRISPR library, designated HIV-Dep, which represents genes that had previously been shown to be required for HIV-1 replication across multiple analysis. The modified CRISPR screen is quite clever in that the guide RNAs are flanked by a packaging signal such that transduced JLAT cells treated with an LRA will produce virions in the supernatant carrying guides, which for genes required for latency reversal will be depleted relative to the genomic knockout library. They validated 20 of the top hits which produced effects in 2 different JLAT cell lines, and ended up focusing on CCNT1, encoding cyclin T1 for downstream analysis. They showed that CCNT1 knockouts prevent reactivation of HIV in response to a variety of latency reversing agents, in cell lines and normal human CD4 cells. Interestingly, RNA seq analysis of CCNT1 knockout cells showed that Cyclin T1 has a minimal effect on host cell genes, and in fact the most significantly affected transcripts in HIV-1 infected cells is the HIV provirus itself, along with several genes known to be upregulated by NFkB and TNF alpha signaling.
This is a very interesting study, which directs important focus towards specificity of the PTEFb complex for HIV-1 expression. The finding that CCNT1 scores as one of the strongest hits in this screen is not surprising, but discovery that the cyclin T1 isoform plays a much more significant role in control of HIV-1 reactivation than for global gene expression, at least in T cells in vitro, is of significant interest. These observations raise a whole series of questions regarding the supposedly "redundant" Cyclin T isoforms, T1 and T2. There are not many studies (or any) that have specifically focused on functional differences for these, this study indicates a detailed analysis is warranted. It would be of particular interest to understand how these different isoforms affect recruitment of PTEFb to cellular promoters. The discussion section might be expanded slightly to include a brief mention of these issues.
Specific comments:
1. Last sentence of the abstract should be modified, and other places throughout the manuscript, for use of the term "redundant". CCNT1 knockout does alter HIV-1 gene expression, but not most cellular genes, and therefore is not redundant, maybe better here to state as "non-essential".
2. Figure 2D, how is the knockout score generated? Western blots are needed to confirm factor depletion.
3. Figure 2D, the data for NTS and AAVS1 are shown as a single bar. These results should be separated as they are different experiments.
4. Section title for 3.2 should read "Cyclin T1 is essential for reactivation from latency".
5. Section 3.2, the statement "CDK9 which binds to Cyclin T1 in order to form the positive transcription elongation factor complex (P-TEFb) is substantially depleted in both cell lines" requires a reference. I am not familiar with this observation.
6. Figure 3, It might be good to include a result showing that the requirement of CCNT1 for HIV-1 transcription involves transactivation by TAT.
7. Figure 3B, the authors measure reactivation of HIV-1 expression using RT activity. Do these results correlate with expression of GFP from the JLAT cell line(s) used here?
8. Figure 4B. The number of genes significantly upregulated upon TNF alpha treatment in these cells seems rather low, as this treatment activates NFkB. How do these results compare to similar analysis of uninfected T cell lines or better yet, normal CD4 T cells? Similarly, T cell activation also results in down-regulation of a significant number of genes, which is not apparent in this Figure. Is treatment with TNF alpha at a concentration that is optimal? (Note, the results in Figure 5 A and B, look more like T cell activation, where there are significant numbers of up and down regulated genes).
9. Relating to Figures 4 C and D, it would be good to provide evidence that these cells are activated as measured by cell surface markers (CD69).
N/A
Author Response
These observations raise a whole series of questions regarding the supposedly "redundant" Cyclin T isoforms, T1 and T2. There are not many studies (or any) that have specifically focused on functional differences for these, this study indicates a detailed analysis is warranted. It would be of particular interest to understand how these different isoforms affect recruitment of PTEFb to cellular promoters. The discussion section might be expanded slightly to include a brief mention of these issues.
--We agree that the results are the start of some very interesting lines of investigation about the roles of CyclinT1 versus CyclinT2. We have expanded the Discussion to better address the sequence differences between the two, although we do not have any insights into the functional relevance of the sequence differences to host cell transcription at this time.
“…Our Latency HIV-CRISPR screen in this study revealed our top hit CCNT1 was able to be knocked out with little effect on cell biology, likely due in part to its paralogs CCNT2 and CCNK. Cyclin T1 and T2 are paralogs that have sequence similarities at the amino acid level in the N-terminal region (81% identity), but the C-terminal domain is more divergent and far less similar (~46% amino acid identity between CCNT1 and CCNT2) (Lin and Peterlin et al. J Biol Chem 2002, Peng and Price et al., Genes and Dev 1998). Cyclin K also is similar in N-terminal domain but has a shorter C-terminal region and therefore is a much smaller protein. One possibility is that there is a sequence in this C-terminal domain that adds specificity for gene regulation in CCNT2 or CCNK that allows for regulation of host genes and recruitment to different cellular promoters. Collectively, our findings suggest that CCNT1’s paralogs are sufficient for transcription elongation of host genes but that CCNT1 is required for transcription of HIV-1 genes.”
Specific comments:
- Last sentence of the abstract should be modified, and other places throughout the manuscript, for use of the term "redundant". CCNT1 knockout does alter HIV-1 gene expression, but not most cellular genes, and therefore is not redundant, maybe better here to state as "non-essential".
--We have made this change.
- Figure 2D, how is the knockout score generated? Western blots are needed to confirm factor depletion.
--We now describe in more detail how the knockout score is generated from genomic sequences. We now add these data to the Supplement including the raw reads and processing to obtain a score of the percentage of genes that are functionally knocked out with large deletions or frameshift mutations. We did not perform Western blots on all of the knockouts, but rather concentrated on the most dramatic of them, the CCNT1 pooled knockout. This Western blot is now included in the Supplement with the knockout scores.
Addition in the methods shown in bold below:
“PCR products were purified using AMPure beads (Beckman Coulter, A63880) or QIAquick PCR clean up kit (Qiagen, 28104) and submitted to Fred Hutch Genomics shared resource for sequencing. Analysis was performed using Inference of CRISPR Edits (ICE) (64). Briefly, ICE analysis works by comparing sanger sequencing from wild-type and CRISPR edited sequences, determining the insertion and deletions from these sequences, and generating knockout scores along with a correlation value as assessments of the knockout.”
- Figure 2D, the data for NTS and AAVS1 are shown as a single bar. These results should be separated as they are different experiments.
--We have now separated NTC and AAVS1 knockouts. There are no statistically significant differences between experiments done with NTC or AAVS1 control. We also reflected this change in the figure caption to represent the number of experiments for each control.
- Section title for 3.2 should read "Cyclin T1 is essential for reactivation from latency".
--We have made this change.
- Section 3.2, the statement "CDK9 which binds to Cyclin T1 in order to form the positive transcription elongation factor complex (P-TEFb) is substantially depleted in both cell lines" requires a reference. I am not familiar with this observation.
--This statement was poorly phrased since we did not mean to imply that CDK9 is depleted. Rather the guides targeting CDK9 were depleted in the screen which implies that CDK9 is also essential for latency reactivation. We thank the reviewer for catching this mistake and have modified the sentence appropriately.
Changes in bold below:
Cyclin T1 (CCNT1) is a well characterized regulator of HIV transcription that binds to the viral protein Tat and TAR (24-26) and was the top hit for both J-Lat models. Additionally, CDK9 which binds to Cyclin T1 in order to form the positive transcription elongation factor complex (P-TEFb) is substantially depleted in the CRISPR screen of both cell lines (Supplemental File 1). In order to explore this hit further across a broader range of LRAs, we generated clonal knockout lines of CCNT1 in the J-Lat 10.6 cell line.
- Figure 3, It might be good to include a result showing that the requirement of CCNT1 for HIV-1 transcription involves transactivation by TAT.
--While we agree that this is an interesting question, it is not possible with the current system because the latent proviruses in the J-Lat cells encode a nearly complete provirus with Tat. We feel that previous literature on the role of pTEFb on Tat-dependent transcription and Tat-independent transcription is sufficiently robust, e.g. Zhang and Rana et al. J Biol Chem. 2000, Barboric and Peterlin et al. Mol Cell 2001, Wei and Jones et al. Cell 1998, Kim and Karn et al. J Mol Biol 2011, and Tyagi and Karn et al. J Virol 2010.
- Figure 3B, the authors measure reactivation of HIV-1 expression using RT activity. Do these results correlate with expression of GFP from the JLAT cell line(s) used here?
--We did not measure GFP in the assays on Jurkat cells in this paper. We believe that measuring RT activity in the supernatant is a much superior assay because it measures the compete virus life-cycle to virus budding and because the RT assay has a much larger dynamic range than GFP expression by flow cytometry. In the past, we have found that these two assays correlate, but the RT assay was more sensitive and more reliable.
- Figure 4B. The number of genes significantly upregulated upon TNF alpha treatment in these cells seems rather low, as this treatment activates NFkB. How do these results compare to similar analysis of uninfected T cell lines or better yet, normal CD4 T cells? Similarly, T cell activation also results in down-regulation of a significant number of genes, which is not apparent in this Figure. Is treatment with TNF alpha at a concentration that is optimal? (Note, the results in Figure 5 A and B, look more like T cell activation, where there are significant numbers of up and down regulated genes).
--We thank the reviewer for this feedback and these are valid concerns. Our goal with TNFalpha treatment was to stimulate reactivation from latency given this has been shown to potently stimulate transcription of latently infected Jurkat cells (Pasquereau and Herbein et al. Viruses 2017). We do note that there is upregulation of several genes that we would anticipate to be upregulated in TNF alpha treated cells, including RelB, PGLYRP4, and BCL3, and the dose of TNFalpha was certainly effective since we see over 100-fold induction of HIV RNA. We are not sure why there are not more genes downregulated in this treatment in Jurkat cells although a few other studies have also noted the relative lack of downregulation of genes relative to upregulation after TNFalpha treatment (albeit, not in T cells and with doses of TNFalpha higher than used in our study) , e.g. Zhou and Williams et al. Nature Oncogene April 2003 and (Bauer and Soliman et al. Oncology Letters 2020). In any case, we believe that the RNAseq data in Figure 5 with primary T cells stimulated through the T cell reception is a more relevant model for the lack of role of CCNT1 in transcriptional control in T cells.
- Relating to Figures 4 C and D, it would be good to provide evidence that these cells are activated as measured by cell surface markers (CD69).
--We did not do this experiment, but as described in point 8, we do see upregulation of genes indicative of a NF-kB response. We feel that looking at activation in primary cells as done in Figure 5 was better in the context of CCNT1 knockout.
Round 2
Reviewer 2 Report
This is a very interesting study, I hope that you will follow up on mechanistic differences between Cyclin T1 and T2